# Safety Evaluation of a Water-Immersed Bridge Against Multiple Hazards via Machine Learning

**Kuo-Wei Liao \*, Fu-Sheng Chien and Rong-Jing Ju**

Department of Bioenvironmental Systems Engineering, National Taiwan University, Taipei 10617, Taiwan
* Correspondence: kliao@ntu.edu.tw; Tel.: +886-2-3366-3447

**Abstract:** A safety analysis process for an immersed bridge is proposed, in which material nonlinear properties, added mass, soil spring, plastic hinge length and property, nonlinear pushover analysis, and time history analysis are included. Outcomes of interest in a bridge analysis, such as displacement ductility, is not an easy task if the authentic model is adopted. The least-squares support-vector machine (LSSVM) is therefore proposed for replacing the analysis model. Water-immersed bridges under varied water depths, scouring depth, and seismic intensity are investigated. The feasibility of using machine learning as a surrogate model is discussed. Results indicate that the trends of fragility curves derived from LSSVM are very similar to those of the authentic model. Therefore, LSSVM is a suitable tool to reduce the computational burden. Analyses also show that stream velocity and pier length are two important factors for the safety of an immersed bridge, and the immersed effect should not be ignored if pier length is long.

**Keywords:** river bridge; multihazards; added mass; fragility analysis; LSSVM

## 1. Introduction

In Taiwan, interweaving rivers and creeks are spread over the entire island. To achieve smooth southbound and northbound traffic flow, river-crossing bridges are prerequisite infrastructure facilities. Because these infrastructure facilities are usually designed with a very long lifespan, it is also required to consider the effects from all aspects when designing these structures. As Taiwan is situated in a subtropical monsoon area, river-related hazards should not be treated lightly because of the abundant rainfall and the striking of typhoons during June–August each year. In the meantime, Taiwan is also located at the interfacing area between the Eurasian and Philippine Sea Plates. Therefore, it sits on the seismic zone. When these plates drift and squeeze with each other, Taiwan will be shaken by earthquakes. As such, impacts to structures as caused by earthquakes should not be ignored, and this is the key point that will be dealt with by each engineer.

Many researchers have focused on building the relationship between earthquake intensities and bridge/building damages. For example, Bazos et al. [1] suggested that fragility curves are a practical tool. Based on the observation at bridge sites, Hsu and Fu [2] found bridges were damaged in different ways in the Chi-Chi earthquake such as unseating span failure, abutment failure, joint failure, substructure damage, footing settlement, and so on. Elnashai et al. [3] developed site-specific ground motions to analyze several typical failure bridge modes and concluded that displacement is a good indicator to measure bridge safety. In addition to the seismic hazard, 44 bridges were damaged from Hurricane Katrina [4]. Andric' and Lu [5] reported that flood-induced damage was the primary damage for bridges in the United States. Similarly, Taiwan faces the same challenge. Liao et al. [6] proposed a bridge safety evaluation process against seismic and flood hazards, in which a scour prediction equation was constructed to measure the flood risk, and the probabilistic seismic-hazard

was adopted to analyze the seismic hazard. To obtain bridge performance, the displacement ductility was used, and nonlinear pushover and time-history analyses were conducted.

The goal of this study is to investigate bridge performance when earthquakes occur during the bridge reinforcing period after being scoured by water. Then the performance of the bridge, including the immersed effect, should be carefully computed. In this case, the fragility curves, obtained from both original and surrogate models, will be employed to learn the hazard-resistant capacity of the bridge. Construction of fragility curves often requires many efforts; to save time, machine learning is often adopted [7]. In addition, due to the superior prediction performance of the least-squares support-vector machine (LSSVM) and several successful studies in safety evaluation or other fields [8–11], this research proposes to use the LSSVM for estimating the ductility capacity of a bridge. LSSVMlab toolbox is an open source program with possibility of having any extension in applications. The formulation of LSSVM is flexible and can be easily incorporated with existing/known information/data due to the parametric model in the primal domain. The aforementioned prior knowledge, such as water level and scouring depth considered here, can be automatically embedded in the dual formulation. Therefore, LSSVM is adopted here. Please note that there are many available machine learning algorithms; investigation or comparison with other algorithms is beyond the scope of the current study. In this article, a bridge located on the QingShui River in Central Taiwan is taken as the target for simulation analysis. The learned displacement ductility from LSSVM will be employed in a bridge safety analysis under all types of scenarios, and then the fragility curves under multiple hazards will be established for calculating the multihazard damage probability. The flowchart of the proposed approach is illustrated in Figure 1. As shown, the following information is obtained according to the design drawing of this bridge: (1) properties of the bridge materials, including the strength of concrete and steel bars, (2) structure layout of the bridge, including the sectional dimensions and reinforcement of bridge deck, cap beam, pier cap, piers and caissons, and so on, And (3) site environment information including the depth of overlying soil, drilling data, 50-year and 100-year flood levels and stream velocities, river sectional width, and so on. After collecting the required information, material nonlinear properties, added mass, soil spring stiffness, and plastic hinge length are calculated in order to establish the bridge model. Once the bridge model is built, nonlinear static and dynamic analyses will be conducted respectively according to the seismic intensity, scouring depth and water level to obtain the displacement ductility. After acquiring the displacement ductility, the LSSVM is trained to estimate the displacement ductility under the considered multiple hazards. The respective occurrence probability can then be calculated, and the fragility curves will be established for different performance levels. The finite element model implemented with SAP2000 Software (Computers and Structures, California, CA, USA) is described in Figure 2.

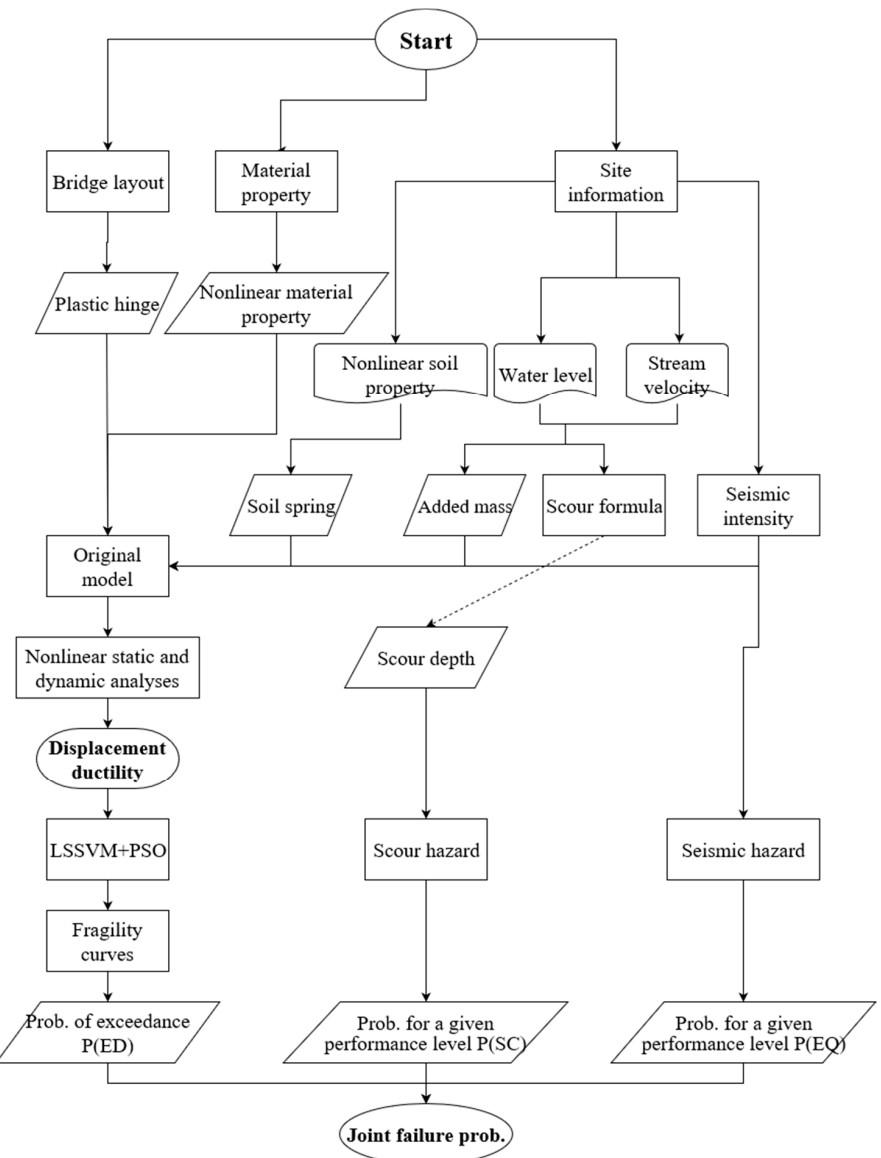

**Figure 1.** Flowchart of the proposed approach. (LSSVM: least-squares support-vector machine, PSO: particle swarm optimization, ED: exceedance, SC: scour depth, and EQ: earthquake).

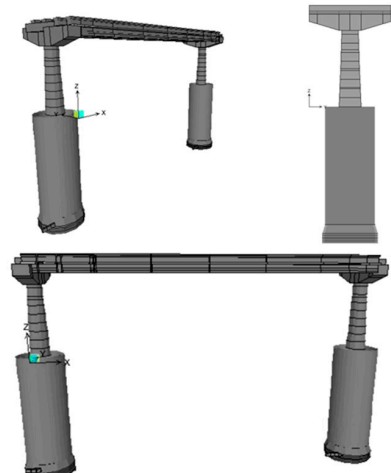

**Figure 2.** The SAP2000 model used in this study.

## 2. Bridge Modeling

In this research, nonlinear behaviors of steel bars were established according to the theory proposed by Priestley et al. [12]. As the steel bars selected were 36 mm in diameter (#11), its limit strain value $\varepsilon_{su}$ was set at 0.09, and the stress–strain hardening initial value $\varepsilon_{sh}$ was set at 0.0115. The corresponding stress–strain curve was drafted as shown in Figure 3. In this research, the bridge pier was configured as a tapered cylindrical column where the top and bottom diameters were 1.8 and 2.4 m, respectively. Because different steel bars were used, the mechanical behaviors of the sectional profile will vary. For this reason, the variable section was divided into 5 sectional profiles in order to calculate the nonlinear mechanics behaviors. Accordingly, 4 tapered cylindrical columns were generated in SAP2000. For example, the topmost portion used the sectional properties of the 1.8 and 1.95 m in diameter, as shown in Figure 4. The interpolated sectional diameters of different height levels were 1.8, 1.95, 2.1, 2.25, and 2.4 m.

The strength of the concrete will be affected by the confining force of stirrups. For this reason, they will be divided into two portions for calculation. One of which is the protection layer, which is the unconfined concrete; and the other is the core concrete, which is a confined layer. For the calculation of confining concrete, the method proposed by Mander et al. [13] was adopted here. Because the steel bars used for this research were 36 mm (#11) in diameter, the limit strains ($\varepsilon_{su}$) of the steel bar and concrete were 0.09 and 0.004, respectively. Note that the limit strain for concrete is 0.003 if American Concrete Institute (ACI) is adopted. After being calculated, the stress–strain curves of the desired sectional profile were obtained (Figures 5 and 6) for confined and unconfined concretes.

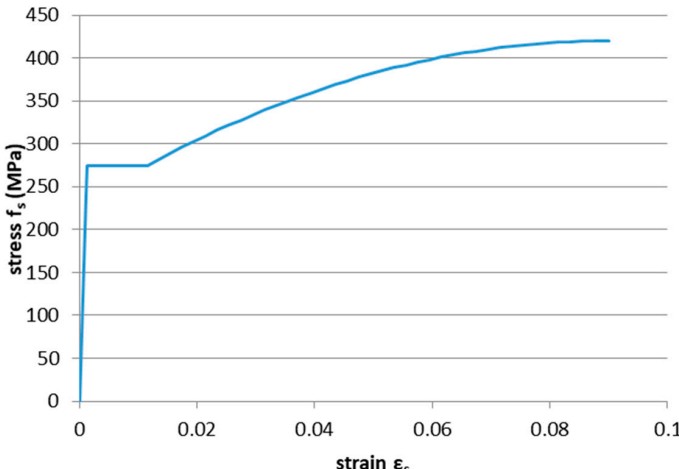

**Figure 3.** Stress–strain curve of a steel bar used in this study.

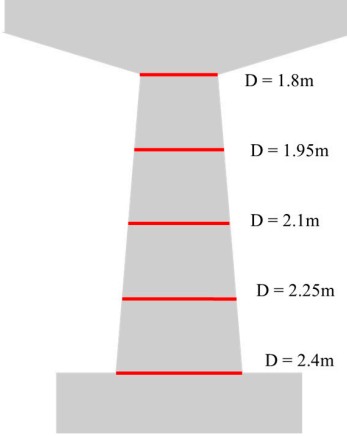

**Figure 4.** Diameter schematic layout of the selected sectional profile.

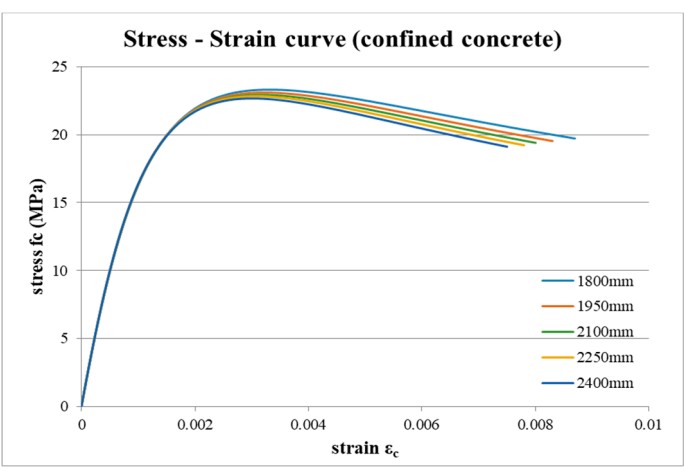

**Figure 5.** Stress–strain curve for confined concrete.

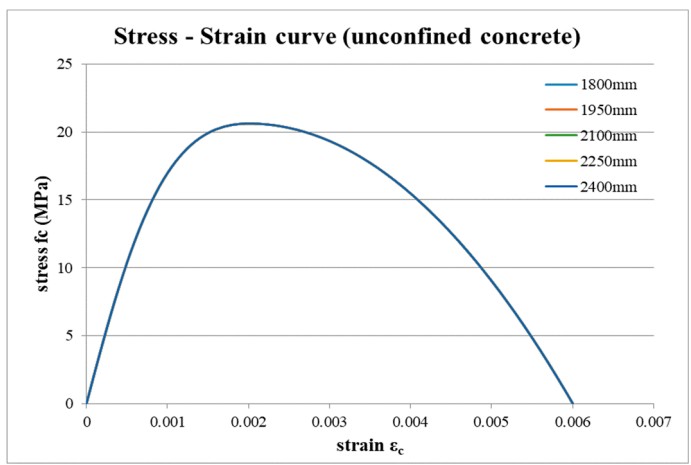

**Figure 6.** Stress–strain curve for unconfined concrete.

Depending on the damage mode, the plastic hinge used for the finite element model had the following properties: flexural damage (corresponding to the bending moment of plastic hinge $M_2$ or $M_3$ in SAP2000), shear damage (corresponding to shear plastic hinge $V_2$ or $V_3$ in SAP2000), and flexural-shear damage (corresponding to the axial force interacting with the bending moment of plastic hinge P-$M_2$-$M_3$ in SAP2000). In SAP2000, the aforementioned three types of plastic hinges can be simulated in three different ways—link, sectional hinge, and fiber—which will vary in accuracy and analysis time. Aviram et al. [14] suggested that the fiber hinge possesses the best accuracy; therefore, it was selected in this article. The fiber hinge resembles the bending moment of the plastic hinge featuring an axial force interaction (P-$M_2$-$M_3$). During analysis, the profile of a target section will affect the analysis time according to the mesh size. For the sake of balancing accuracy and time, the trial-and-error method was used for deciding the mesh size. The finalized meshes for pier and caisson are described in Figures 7 and 8.

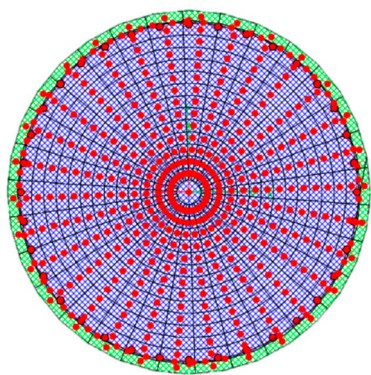

**Figure 7.** Sectional mesh of pier in this study.

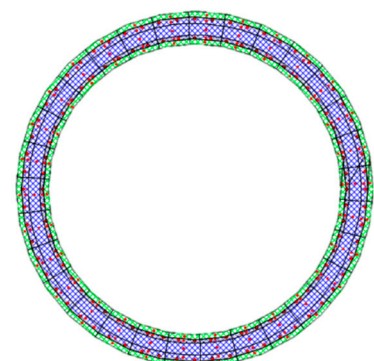

**Figure 8.** Sectional mesh of caisson in this study.

This article used the formula defined by Paulay and Priestley [15] to calculate the length ($L_p$) of plastic hinge as below:

$$L_p = 0.08h + 0.022d_{bl}\sigma_y \ (\text{mm}), \tag{1}$$

where $h$ means pier length (mm), $d_{bl}$ means diameter of the longitudinal steel bar (mm), and $\sigma_y$ means steel bar yielding stress (MPa). Soil resistance around the foundation was simulated by a spring element, in which the front and bottom sides of the foundation were separately considered. During simulation, the soil spring was assumed as a bilinear spring, in which the maximum value did not exceed the passive soil pressure. The soil springs on the front side were deployed at 1 m intervals, and the soil spring was removed if scour occurred.

## 3. Effect of Immersing in Water

As the natural frequency of the river-crossing bridge will vary according to the water level, the influence of immersing the structure in water should be discussed. The immersing effect is considered using the Morison equation [16].

The Morison equation is a semiempirical equation for calculating the force acted on a body under oscillatory flow. There are two parts in the Morison equation, an inertia force that depends on the acceleration of the surrounding flow and a drag force that depends on the velocity of the surrounding flow. Two empirical hydrodynamic coefficients are needed when the Morison equation is adopted: an inertia coefficient ($C_m$) and a drag coefficient ($C_d$). These two coefficients depend on the Keulegan–Carpenter number, Reynolds number, and surface roughness [17]. Yang and Li [18] proposed the added mass ratio method (AMRM) according to the Morison equation, by which a rational method was inducted for calculating the added mass and was verified with the finite element method. In this article, AMRM was adopted, and the added mass ($\Delta m_{cir}$) calculation equation is described as below:

$$\Delta m_{cir} = \rho_{con} \cdot \frac{\pi D^2}{4} \cdot p_{cir}(H, D), \tag{2}$$

where $\rho_{con}$ represents the density of concrete, and the added mass ratio $p_{cir}(H, D)$ of each cylinder section is calculated as below:

$$p_{cir}(H, D) = [0.0133 \ln(H) - 0.112] \times \ln(D) + 0.0002H + 0.4, \tag{3}$$

in which $H$ means the cylinder height, and $D$ means the diameter of the cylinder section. Because the bridge piers of this study were tapered, their added mass was calculated from the pier bottom for every 1 m interval, and then the calculated added mass will be applied to each point, as shown in Figure 9.

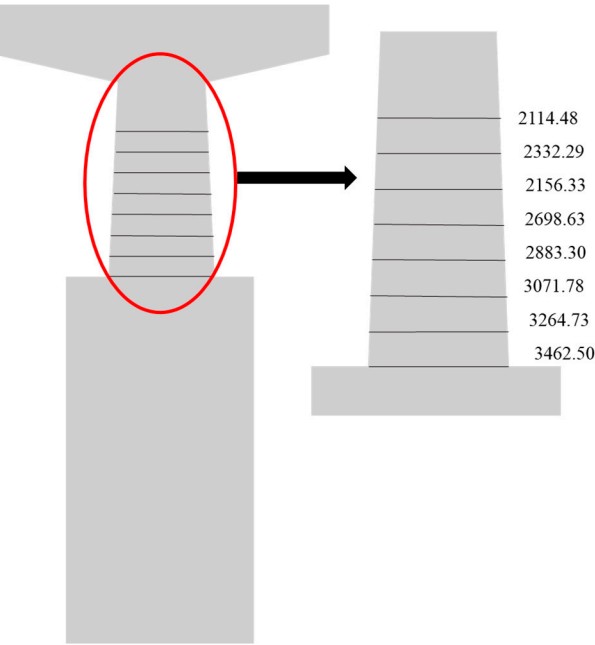

**Figure 9.** Illustration of the calculated added mass (kg).

## 4. Simulation of Water Depth, Stream Velocity, and Scouring Depth

When calculating bridge scouring depth, two parameters must be obtained: water depth and stream velocity. Because the water depth and the stream velocity should be observed over time, normally they are calculated through simulation. Presuming that the mean value of water depth and stream velocity in different return years can be regarded as a log-normal distribution with three parameters, the water depth and stream velocity of the same return year can also be regarded as a log-normal distribution. Based on the above assumptions and the information revealed in the design drawing (i.e., the discharge rate for return periods of 50 and 100 years), the mean value of water depth $y_{200}$ and stream velocity $V_{200}$ for a 200-year return period were obtained. Furthermore, the coefficients of variation of water depth and stream velocity were presumed identical under different return periods. In this way, we may acquire the probability distributions of water depth and stream velocity for 50, 100, and 200 year return periods, as per the detailed calculation method. Provided below is the hydrological frequency equation:

$$\hat{x} = \bar{x} + K_T \cdot S, \ K_T = \frac{\exp(\sigma_y \cdot t - \frac{\sigma_y^2}{2}) - 1}{[\exp(\sigma_y^2) - 1]^{1/2}}, \tag{4}$$

where $\bar{x}$ and $S$ represent mean value and standard deviation of the sample, respectively. $K_T$ represents frequency factor and is calculated below;

$$\sigma_y = [\ln(z^2 + 1)]^{1/2}; \tag{5}$$

$$z = \frac{1 - w^{2/3}}{w^{1/3}};\tag{6}$$

$$w = \frac{-C_s + \sqrt{C_s{}^2 + 4}}{2};\tag{7}$$

where $C_s$ means the skewness. Parameter $t$ in Equation (4) can be acquired by the formula below:

$$\begin{cases} t = -W + \frac{C_0 + C_1 W + C_2 W^2}{1 + d_1 W + d_2 W^2 + d_3 W^3}, \ W = \sqrt{-2\ln(1 - P)}, \ \text{if } P > 0.5 \\ t = W - \frac{C_0 + C_1 W + C_2 W^2}{1 + d_1 W + d_2 W^2 + d_3 W^3}, \ W = \sqrt{-2\ln(P)}, \ \text{if } P \leq 0.5 \end{cases},\tag{8}$$

where $P$ is the inverse of the return period, $C_0 = 2.515517$, $C_1 = 0.802853$, $C_2 = 0.010328$, $d_1 = 1.432788$, $d_2 = 0.189269$, and $d_3 = 0.001308$. Through calculations with the aforesaid equation, the average water depth $y_{200}$ as 5.9525 m and flowing speed $V_{200}$ as 2.407 m/s for the 200 year regression period were obtained. To calculate scour depth, especially to obtain the uncertainty of scour, a simulation of water level and stream velocity was necessary. Liao et al. [19] indicated that the correlation coefficient between water depth and stream velocity was about 0.92. Based on this, a Copula function was used in this research for generating random samples for water depth and stream velocity. After acquiring the water depths and stream velocities for 50, 100, and 200 year return periods, they were revolved in the formula proposed by Melville and Coleman [20], respectively, in order to acquire the corresponding distribution of scouring depth, as shown below.

$$d_s = K_s K_t K_G K_\theta K_{yb} K_I K_d.\tag{9}$$

In the formula, $d_s$ represents the scour depth (m), $K_s$ means foundation shape factor, $K_t$ means time factor, $K_G$ means approach channel geometry factor, $K_\theta$ means foundation alignment factor, $K_{yb}$ means foundation size factor, $K_I$ means flow intensity factor, and $K_d$ means sediment size factor. Among these factors, $K_{yb}$ poses the most significant impact. Detailed calculations of $K_{yb}$ in this article is provided below.

Based on the classification of caisson width and elevation of river bed, a pier-equivalent width was derived for calculating the bridge scour [21]. Liao [22] utilized an optimization approach to update the parameter weights for the pier-equivalent width formula proposed by Melville and Raudkivi [21], as displayed below.

$$b_e = b\left(\frac{-0.002742y + 0.000625Y}{-0.002742y - 0.000122b_{pc}}\right) + b_{pc}\left(\frac{-0.000075b_{pc} + 0.000388Y}{-0.000075b_{pc} - 0.001703y}\right),\tag{10}$$

where $b_e$ represents the equivalent width of bridge, $b$ means the width of pier, $b_{pc}$ means the width of caisson, $y$ means the depth from water face to riverbed, and $Y$ means the elevation from caisson top to riverbed. The foundation size factor $K_{yb}$ will vary according to the ratio between bridge equivalent width $b_e$ and water depth and its calculation formula is as below:

$$\begin{cases} K_{yb} = 2.4b_e & \frac{b_e}{y} < 0.7 \\ K_{yb} = 2\sqrt{yb_e} & 0.7 < \frac{b_e}{y} < 5 \\ K_{yb} = 4.5y & \frac{b_e}{y} > 5 \end{cases}.\tag{11}$$

Based on Liao's updated formula, the probability of exceedance and occurrence for the scour distribution for the 100 year return period are shown in Figures 10 and 11, respectively.

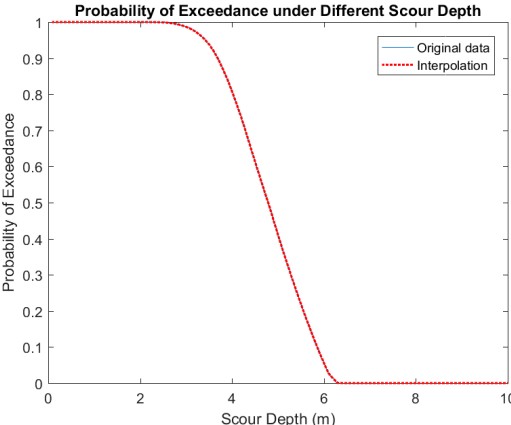

**Figure 10.** Scouring depth exceeding probability for the 100 year return period.

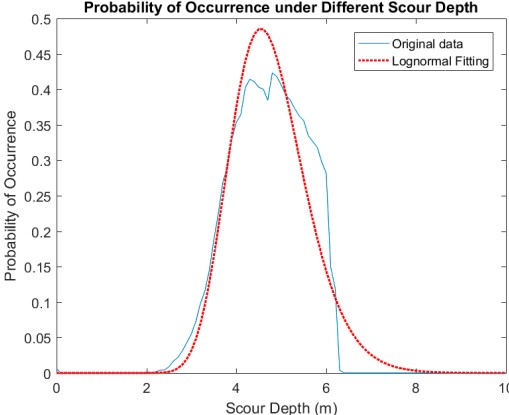

**Figure 11.** Scouring depth probability density function for the 100 year return period.

## 5. Seismic Hazard Analysis

Based on the seismic design code in Taiwan, horizontal spectrum acceleration coefficients $S_{DS}$ and $S_{Dm}$ were obtained for the investigated bridge site. Based on the short period peak ground motion (PGA) value, the EPA (effective peak ground acceleration) of the design earthquake (about 475 years return period) and the maximum consideration earthquake (about 2500 years return period) can be calculated as below.

$$EPA_D = 0.4S_{DS} = 0.4 \cdot F_a \cdot S_S^D \cdot N_A = 0.4 \times 1.0 \times 0.8 \times 1.16 = 0.3712(g) = PGA_D; \tag{12}$$

$$EPA_M = 0.4S_{DM} = 0.4 \cdot F_v \cdot S_S^M \cdot N_A = 0.4 \times 1.0 \times 1.0 \times 1.16 = 0.464(g) = PGA_M. \tag{13}$$

This study presumed that the bridge lifespan was set at 50 years. On this basis, the exceedance probability (EP) of an earthquake, in magnitude greater than or equal to 475 years and with 2500 years of reoccurrence period that has occurred at least once in such duration, shall be calculated as below:

$$P(k|k \geq 1, 50) = 1 - e^{-\frac{50}{T}} = 1 - e^{-\frac{50}{475}} \cong 10\%; \tag{14}$$

$$P(k|k \geq 1, 50) = 1 - e^{-\frac{50}{T}} = 1 - e^{-\frac{50}{2500}} \cong 2\%. \tag{15}$$

Presuming that the probability of magnitude greater than or equal to *M*, which is indicated by Equations (14) and (15), during the lifespan follows a log-normal distribution, then the exceedance probability distribution and the occurrence probability distribution of the bridge during the lifespan can be acquired as indicated in Figures 12 and 13.

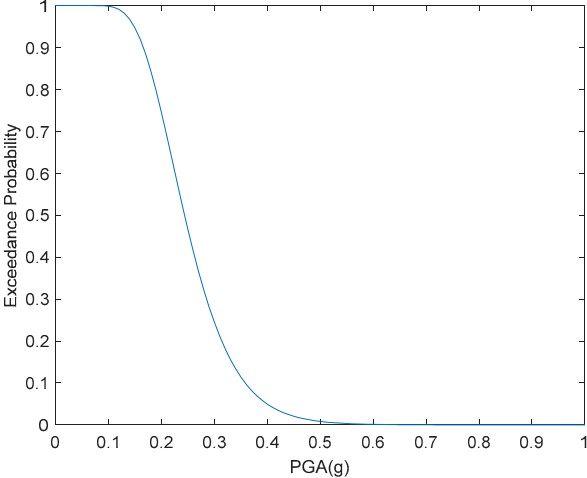

**Figure 12.** Exceedance probability distribution of the bridge considered having a 50 year of lifespan.

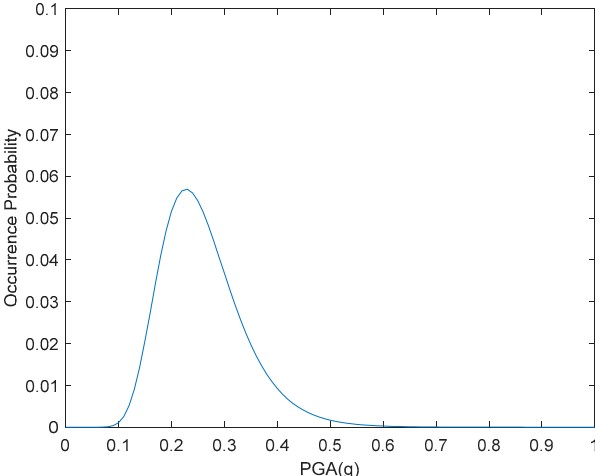

**Figure 13.** Occurrence probability distribution of the bridge considered having a 50 year of lifespan.

## 6. Fragility Curve Analysis

A fragility curve is used to indicate the failure probability of the structure under different levels of hazard, and the fragility curve is mainly affected by the demand and the capacity of the bridge. The demand means the displacement ductility ($R_D$) analyzed by simulating a scoured bridge under specific earthquake magnitude. As for the capacity ($R_C$), it is calculated according to the definition proposed by Federal Emergency Management Agency (FEMA, HAZUS-MH/MR3, [23]), where $1 < R_c < 2$ means minor damage, $2 < R_c < 4$ means medium damage, $4 < R_c < 7$ means major damage, and $R_c > 7$ can be considered that the structure completely collapsed. When the demand of displacement ductility is bigger than the capacity, it can be assumed that the bridge is damaged. Here, the lower bounds of each performance level (e.g., 1, 2, 4, and 7) were used as the threshold values. The displacement ductility demand ($R_D$) of the bridge is the ratio between the ultimate displacement ($\Delta_u$) and yielding displacement ($\Delta_y$), that is, $R_D = \frac{\Delta_u}{\Delta_y}$. Assuming the pier of the bridge serves as the primary damage mechanism, in this case, the aforementioned term of "displacement" refers to the relative displacement between pier top and pier bottom. In view that it would be impossible to acquire an accurate ultimate displacement of the bridge by a nonlinear static analysis method, such as a pushover analysis, the ultimate displacement was therefore calculated by a nonlinear time history analysis, where the maximum value of the bridge displacement duration excitation is the ultimate displacement.

The yielding displacement ($\Delta_y$) is another key in determining the displacement ductility demand ($R_D$), where $\Delta_y$ is acquired from a nonlinear pushover analysis. Because the pushover curve is irregular in nature, the method of acquiring $\Delta_y$ has always been a subject discussed by all sectors. In this study, the equal energy method proposed by Sung et al. [24] was adopted to calculate the yielding displacement, and it will be explained in the following three steps.

First, acquire area ($A_1$) under the pushover curve. Then, draft the tangent lines on the origin and the ending points of the curve, where the corresponding slopes are considered as the initial stiffness ($K_1$) and yielding stiffness ($K_2$). Accordingly, the intersection point ($P_1$) is also obtained, as per Figure 14a below. Next, rotate the initial stiffness ($K_1$) line from its original point clockwise until area ($A_2$), which is enclosed by the rotating and $K_2$ lines, becomes equal to area ($A_1$). Then, we have intersection point ($P_2$), as per Figure 14b below. Likewise, rotate the yielding stiffness ($K_2$) line from its ending point counter clockwise until area ($A_3$), which is enclosed by the rotating and $K_1$ lines, becomes equal to area ($A_1$). Then, we have intersection point ($P_3$), as per Figure 14c below. Finally, a line which is orthogonal to the line of $\overline{P_2 P_3}$ and passes through the $P_3$ point can be drafted in which the intersection point is the yielding point ($P_y$), and the corresponding displacement is the yielding displacement ($\Delta_y$), as per Figure 14d below. Indicated in Figure 15 is an example of the yielding displacement ($\Delta_y$) calculated from a pushover curve with 0 m of water depth and scouring depth.

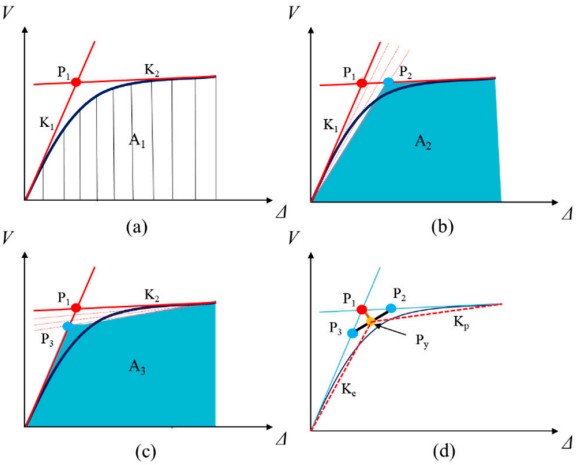

**Figure 14.** Schematic illustration of acquiring the yielding point in this study ($V$ is the base shear and $\Delta$ is the relative displacement between pier top and bottom). (**a**) Initial and yielding stiffness. (**b**) Illustration of $P_2$ point. (**c**) Illustration of $P_3$ point. (**d**) Illustration of yielding point.

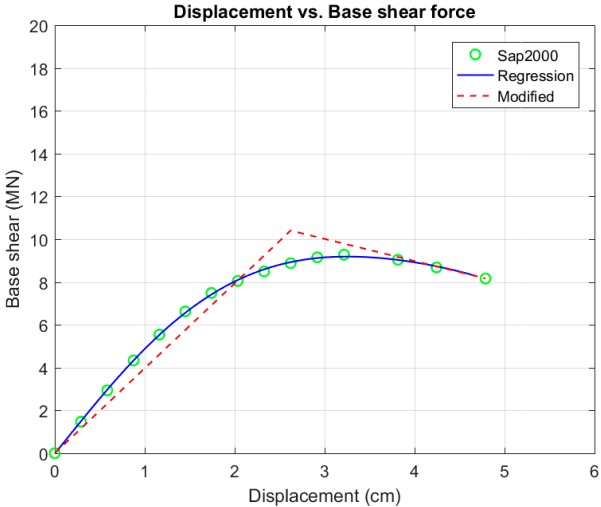

**Figure 15.** Example of yielding displacement for a pushover curve.

After acquiring the displacement ductility demand ($R_D$), the failure probability can be analyzed in order to obtain the fragility curve. When analyzing the failure probability, a limit state needs to be defined, in which $PE = \frac{R_C}{R_D} - 1$ was used in this study, indicating that when $PE > 0$, the displacement ductility capacity ($R_C$) of a bridge is bigger than displacement ductility demand ($R_D$), and the considered bridge is under safe conditions. Otherwise, it means that the bridge will be subject to damage. As such, the failure probability, when a bridge is subject to seismic force PGA = *x*, can be expressed by the following formula:

$$P(PE \leq 0 | \text{PGA} = x) = P(\frac{R_c}{R_D} \leq 1 | \text{PGA} = x) = P(R \leq 1 | \text{PGA} = x), \tag{16}$$

in which $R$ is $\frac{R_C}{R_D}$. Assuming $R_C$ and $D_C$ are log-normal distributions, then the formula used for calculating the failure probability can be revised as:

$$1 - \Phi(\frac{\mu_{\ln(R_C)} - \mu_{\ln(R_D)}}{\sqrt{\sigma^2_{ln(R_C)} + \sigma^2_{ln(R_D)}}}), \tag{17}$$

where $\mu_{\ln(R_C)}$ refers to the mean capacity and $\sigma_{ln(R_C)}$ refers to standard deviation of the capacity, and $\mu_{\ln(R_D)}$ refers to the mean demand and $\sigma_{ln(R_D)}$ refers to standard deviation of the demand. Based on the research report [25] proposed by National Center for Research on Earthquake Engineering (NCREE) of Taiwan, the standard deviations of capacity of the respective performance level (e.g., slight, moderate, extensive, and collapse) are 0.5, 0.45, 0.4, and 0.4 respectively. $\Phi(\cdot)$ represents the standard normal cumulative distribution function. The required mean value and standard deviation of demand were obtained from the regression of power law. To do so, 3 sets of data were used: the displacement ductility demands ($R_D$) for the 30 year return, 475 year return, and 2500 year return periods, respectively. Each set contained 7 structural responses from time history analyses using 7 earthquake excitations. Listed below are the formulas used to calculate the PGA, mean, and standard deviation displacement ductility demand.

$$\mu_{R_D} = a(PGA)^b, \tag{18}$$

$$\sigma_{R_D} = c(PGA)^f, \tag{19}$$

where *a*, *b*, *c*, and *d* represent the regression parameters. Note that at least 7 ground motions should be used if a time history analysis is used to measure the structural performance (LRFD seismic bridge design, AASTHO 2007). In addition, the "response-spectrum-compatible" time histories were used here. A response-spectrum-compatible time history refers to the response spectrum of the selected earthquakes falling between 0.2 and 1.5 T (T is the fundamental period), which may not be less than 90% of the corresponding design spectral acceleration for a damping ratio of 5%. In addition, the average value of the response spectrum within the designated period range may not be less than the average value of the corresponding design spectral accelerations. The 7 ground motions selected here were therefore converted into response-spectrum-compatible data for return periods of 30, 475, and 2500 years.

## 7. Least-Squares Support-Vector Machine (LSSVM)

The aforesaid fragility curve was drafted according to the many results of structural analyses, and it is a time-consuming process. For this reason, LSSVM was used to construct a surrogate model and to rapidly establish the fragility curve for evaluating the performance of a bridge. The concept of LSSVM is to consider the key essential parameters as input vectors for a machine, and together with given outputs, the machine can be trained to learn in order to estimate the result through the learning process. The parameters (training input) considered in this article were the PGA, water level, and scouring depth. The displacement ductility demand ($R_D$) obtained from the structural analysis was used as the output data during the training. The LSSVM is briefly introduced below.

SVM is a classifier that is able to solve a nonlinear problem using convex quadratic programs (QPs), as shown in Equation (20).

$$\underset{w,b,\xi}{\text{minimize}} \; \tfrac{1}{2}w^T w + c \sum_{k=1}^{N} \xi_k$$
$$\text{Subject to} \begin{cases} y_k(w^T K(x_i) + b) \geq 1 - \xi_k \\ \xi_k \geq 0, \; i = 1, 2, \ldots, N \end{cases} \tag{20}$$

in which $y_k$ represents the class and $(w^T K(x_i) + b)$ indicates the classifier, $w$ is a vector of weights that are orthogonal to the hyper-plane, $c$ is a constant number that is greater than zero, and $\xi_k$ is the slack variable. When $\xi_k$ is greater than one, this indicates that the $k$-th inequality is violated. $N$ is the number of data, and $K$ is the kernel function, in which a Gaussian radial basis function (RBF) is one of the common kernels and was adopted here, as shown in Equation (21).

$$K(X, X_i) = e^{-\sigma(\|X - X_i\|)^2}, \tag{21}$$

in which vector $X$ is an input, $\sigma$ represents the kernel function parameter, and $X_i$ is the support vector. The least-square support-vector machine (LSSVM, Suykens and Vandewalle, [26]) does not attempt to solve the QP problem. LSSVM actually tries to solve a system of linear equations after altering the SVM via introducing the error variable ($\varepsilon$), as described in Equation (22).

$$\min \tfrac{1}{2}w^T w + \tfrac{\gamma}{2} \sum_{k=1}^{N} \varepsilon_k^2$$
$$\text{s.t. } y_k(w \cdot K(x_k) + b) = 1 - \varepsilon_k, \; k = 1, \ldots, n \tag{22}$$

in which $\gamma$ is a constant number. It is seen that two modifications, equality constraints and a squared error variable, leads to solving a set of linear equations in LSSVM.

When training the LSSVM, the training samples are separated into two portions. In the present case, 525 analysis data points previously obtained were split by an 8:2 ratio, randomly, in which 80% was used as the training samples and 20% was used to compare the estimate result. During training, these samples were separated into 10 groups for cross-validation in order to improve its accuracy and avoid overfitting. To implement the LSSVM in a computer, this study utilized the LSSVMlab toolbox provided by Suykens et al., which contains Matlab/C implementations for a number of LSSVM algorithms. To measure the performance of the trained LSSVM, 4 sets of indices were used: mean absolute error (MAE), mean absolute percentage error (MAPE), root-mean-square error (RMSE), and coefficient of determination ($R^2$). The hyper-parameters ($\alpha$ and $\beta$) of LSSVM are determined by particle swarm optimization (PSO).

The joint-failure probability is assumed to be the product of three probabilities [27]: the probability of seismic hazard, the probability of scour hazard, and bridge failure probability for a given limit state. Accordingly, the joint failure probability can be expressed as:

$$\left(P_f\right)_{ijk} = P(SC_i) P\left(EQ_j\right) P(DS_k), \tag{23}$$

where $P(SC_i)$ is the probability of occurrence of a given scour depth ($i$), acquired from the scouring hazard analysis; $P\left(EQ_j\right)$ is the probability of occurrence of a certain PGA ($j$), acquired from seismic hazard analysis; and $P(DS_k)$ is the exceedance probability of a bridge exceeding a certain performance level ($k$), acquired from time history analyses.

The hyper-parameters ($\alpha$ and $\beta$) of LSSVM are determined by PSO. PSO is briefly introduced below. To find the optimum, PSO mimics the flight of a flock of birds and simplifies the social behavior with integrated individuals. PSO is a popular algorithm that has been used in many optimization problems [28]. In PSO, a population/swarm means a set of particles, and their corresponding position

is one possible solution. The optimal direction for the next step of each particle is calculated based on the velocity according to best experiences of the particle itself and the swarm. The position, $X_i^d$, and the velocity, $V_i^d$, of the *i*-th particle in a population is recalculated as shown below.

$$V_{i+1}^d = \omega \cdot V_i^d + c_1 \cdot r_1 \cdot (pbest_i^d - X_i^d) + c_2 \cdot r_2 \cdot (gbest_i^d - X_i^d), \tag{24}$$

$$X_{i+1}^d = X_i^d + V_{i+1}^d, \tag{25}$$

where *pbest* is the best solution up to the current iteration of the particle itself, *gbest* is the best solution among *pbests* up to the current iteration, $r_i$ is a random number between 0 and 1, $c_1$ and $c_2$ are acceleration parameters for cognitive and social behaviors, and $\omega$ is the inertia weight. Deb's rules is one of the popular methods to handle the constraints, and it was adopted in the current study.

## 8. Results and Discussion

Indicated in Figure 16 is the fragility curve of the respective scouring depth under the same performance level. As shown, the failure probability was not proportional to the scour depth. In addition, the performance level of slightly damaged had the least failure probability. To show this trend in a clearer way, Figure 17 denotes the fragility curve of the respective performance level under the same scouring depth and water level. As shown, the fragility curves indicate that the deeper the scour depth, the higher the failure probability, and the lower the performance level, the higher the failure probability. Such conclusion is consistent with the concept held by general public reasoning that the strength of a bridge foundation will degrade after being scoured and eroded. However, varied water depth tends to affect the failure probability of the bridge. Indicated in Figure 18 are the fragility curves of each water depth under the same scouring depth (0 m) and the same performance level. Taking the slight damage level for example, the failure probability increased from 43% to 56% if the water depth increased from 0 m to 8 m at PGA of 0.2 g. As seen, the influence of water depth was on the nonconservation side, and the influence of water depth should be considered, especially, when longer piers are used.

Indicated in Table 1 is the result of errors and relative coefficients obtained during LSSVM training for the case when the scour depth was 0 m. It is shown that the prediction performance was not perfect, but it was acceptable. To be more specific, Lewis [29] suggested that if the MAPE is less than 10%, the prediction performance is considered as highly accurate. If the MAPE is between 11% to 20%, the prediction performance is good. After completing the LSSVM training, the corresponding result can be obtained by inputting the input parameters, such as water depth and PGA, in the trained machine to compute the fragility curve. Indicated in Figure 19 is the fragility curve under different scouring depths when the water depth was 3.5 m. Note that the purpose of LSSVM is to provide results that are not calculated by an authentic model such as SAP2000. For example, scour depths shown in Figure 19 were 3.25, 5, 6.25, and 7.5 m. By comparing Figures 16 and 19, it is learned that the LSSVM will provide a similar kind of failure probability trend. When viewing from a moderate limit state, the failure probability between 4 and 8 m of scouring depth, as indicated in Figure 16, was 0.50–0.70 (at 0.3 g), and that in Figure 19 was 0.52–0.72 (at 0.3 g) between 3.25 and 7.5 m of scouring depth. This indicated that the failure probability slightly increased when the immersion depth increased from 0 to 3.5 m. Similar conclusions can be inferred under other performance levels indicated in Figures 16 and 19.

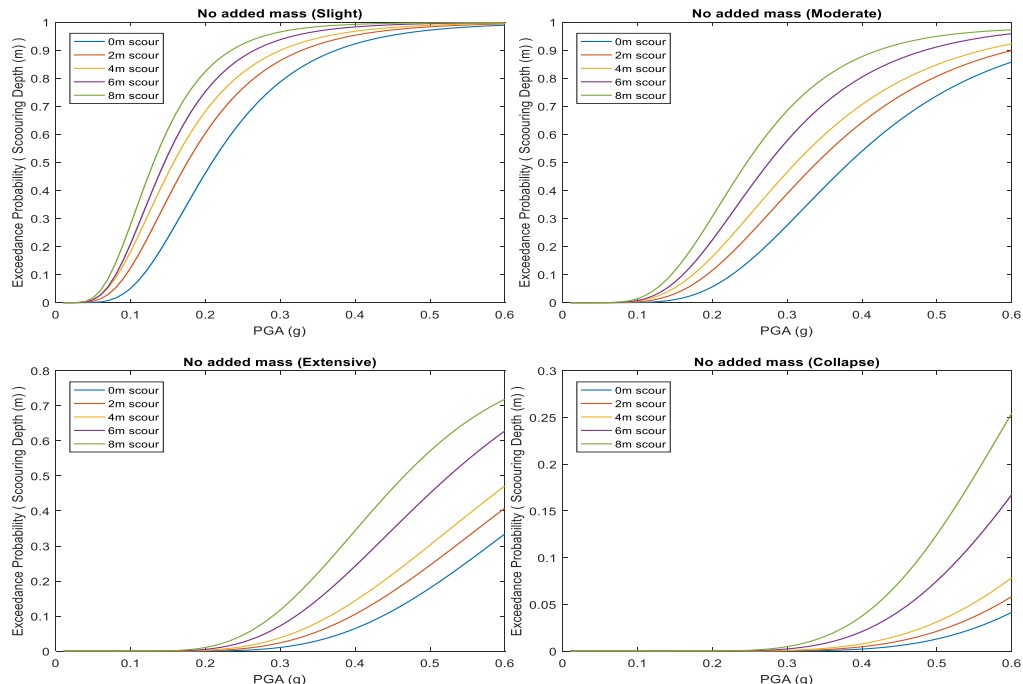

**Figure 16.** Fragility curves of each scouring depth under 0 m bridge water level and the same performance level.

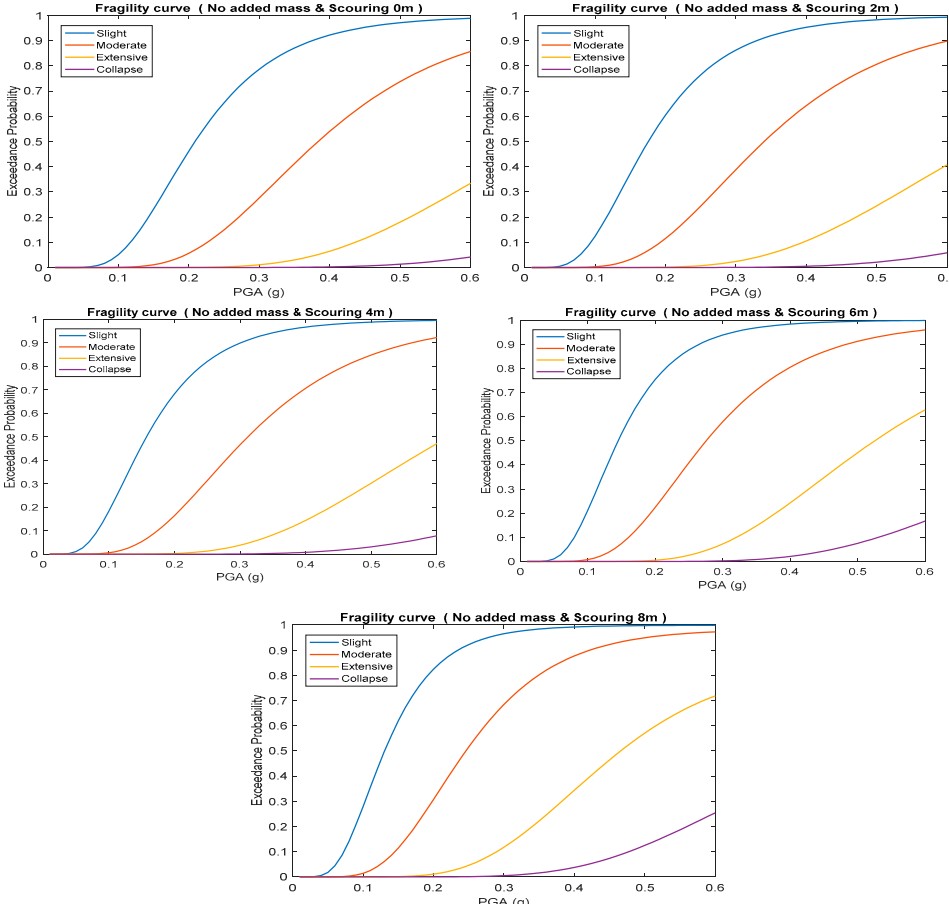

**Figure 17.** Fragility curves of each performance level under 0 m bridge water level and the same scouring depth.

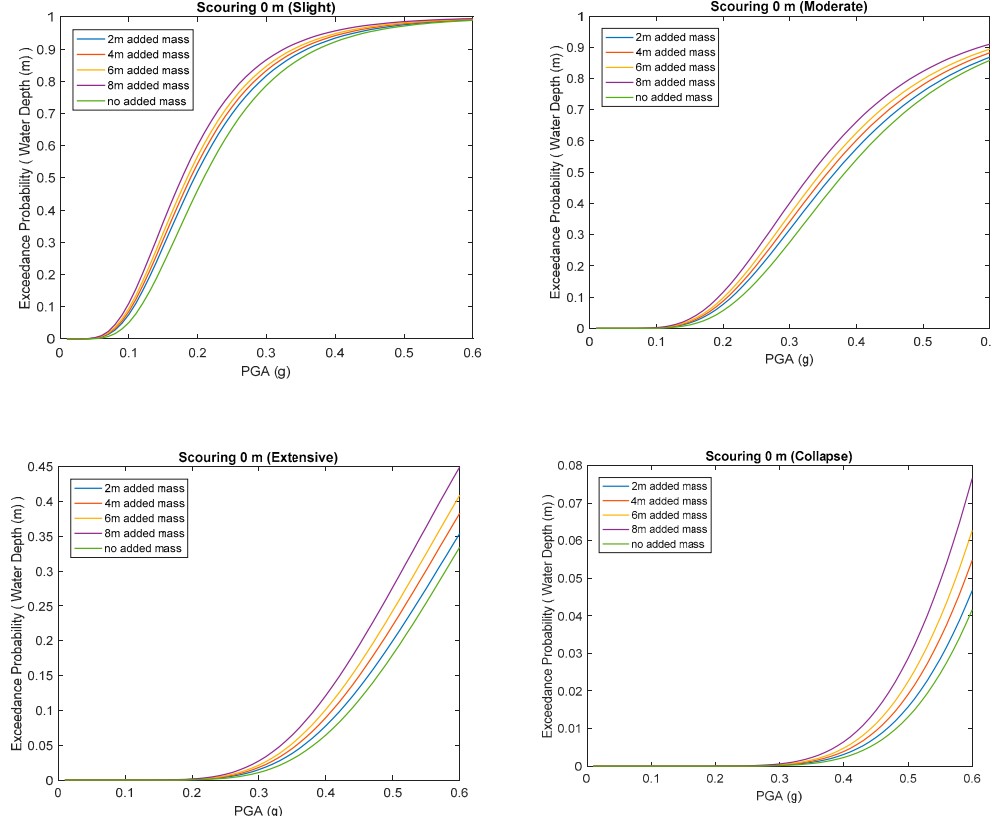

**Figure 18.** Fragility curves of each water level under 0 m scouring depth and the same performance level.

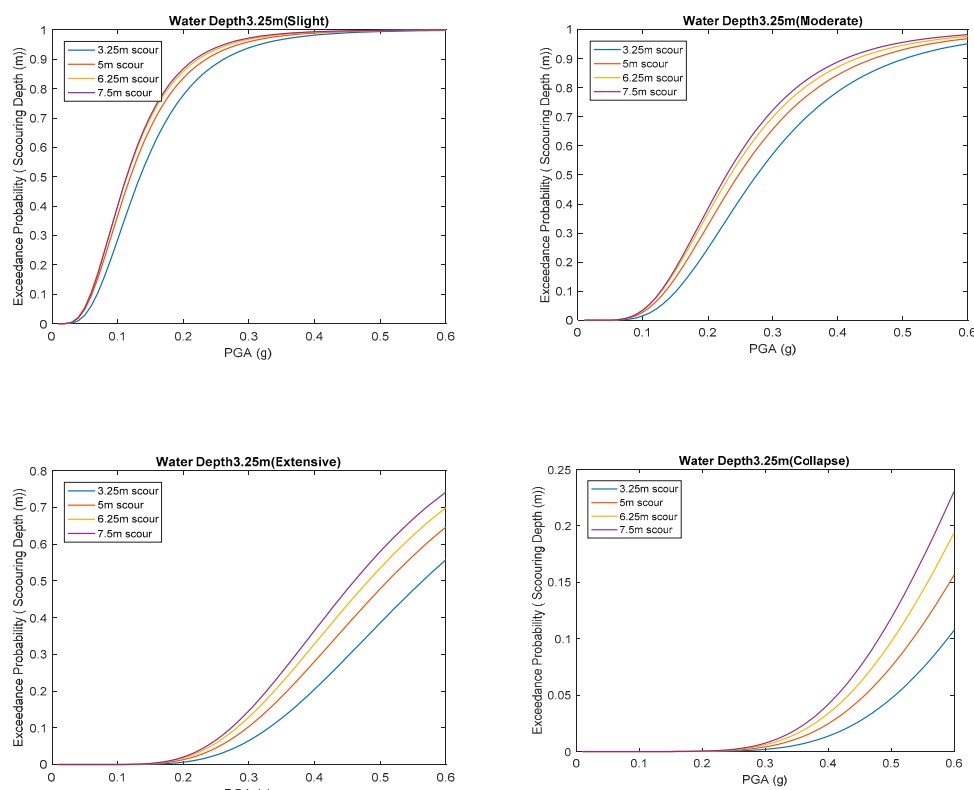

**Figure 19.** Fragility curves of each scoring depth under the same water level and same performance level using LSSVM.

**Table 1.** Result values from the training.

| Parameter | Value |
|---|---|
| Mean absolute error (MAE) | 0.2167 |
| Mean absolute percentage error (MAPE) | 10.76 |
| Root-mean-square error (RMSE) | 0.3209 |
| $R^2$ | 0.9053 |
| $R$ | 0.9582 |

Through the aforesaid comparisons, we can learn that the fragility curves obtained from the fragility curve established by LSSVM and structural analysis are very close. The failure probability of multiple hazards can be calculated according to the scouring hazard curve (Figure 10), seismic hazard curve (Figure 12), and bridge conditional failure probability obtained from nonlinear pushover and time history analyses, given these specific hazards. A direct comparison of using two different approaches (authentic and LSSVM) are displayed as shown in Figure 20. It is seen that the trends of two approaches were similar. The greater the scour depth, the bigger the difference. Please note that the interpolated data between authentic ones may have a higher/lower exceedance probability (EP) than that of the authentic one, as shown in Figure 20. The proposed surrogate model is applied to Equation (23) only when the authentic model is not available. That is, if EP from the authentic model is unavailable, interpolated data from LSSVM was adopted to build plots in Figure 21. The main purpose of utilizing LSSVM for fragility analysis is to reduce the computational cost and retain the correct trends for engineering preliminary analysis. A failure decision for a given bridge depends not only on the failure probability but also on other factors such as the safety threshold, PGA design, scour depth, and performance level. Taking the bridge in Figure 20 for example, based on the bridge site, the PGA design was 0.24 g, assuming that the safety threshold was 0.00135, which is equivalent to a common reliability index (i.e., $\beta = 3$) used for infrastructure. Under these conditions, it was found that among four different scour depths with collapse performance limit states considered, the case of 8 m scour can be considered as a false negative event. Thus, if the predicted fragility is used accompanied with other factors. such as reliability target, scour depth, and PGA, for bridge safety judgement, caution should be taken. If computation time is an issue, LSSVM provides an alternative to shorten the analysis time. The effectiveness of the LSSVM approach depends on the complexity of a given bridge. If the authentic approach is used, many pre-analyses such as material nonlinear properties, added mass, soil spring, plastic hinge length and property, nonlinear pushover analysis, and time history analysis are needed to build the fragility curve. Taking nonlinear time history analysis in this study for example, for each seismic intensity (e.g., the 475 year return period), seven analyses are needed. Such analyses often need several days or weeks to complete. If LSSVM is adopted, one can acquire the displacement within a few minutes. Indicated in Figure 21 is the joint failure probability distribution subjected to 100 year return period scouring when the bridge is immersed underwater at a depth of 4 m.

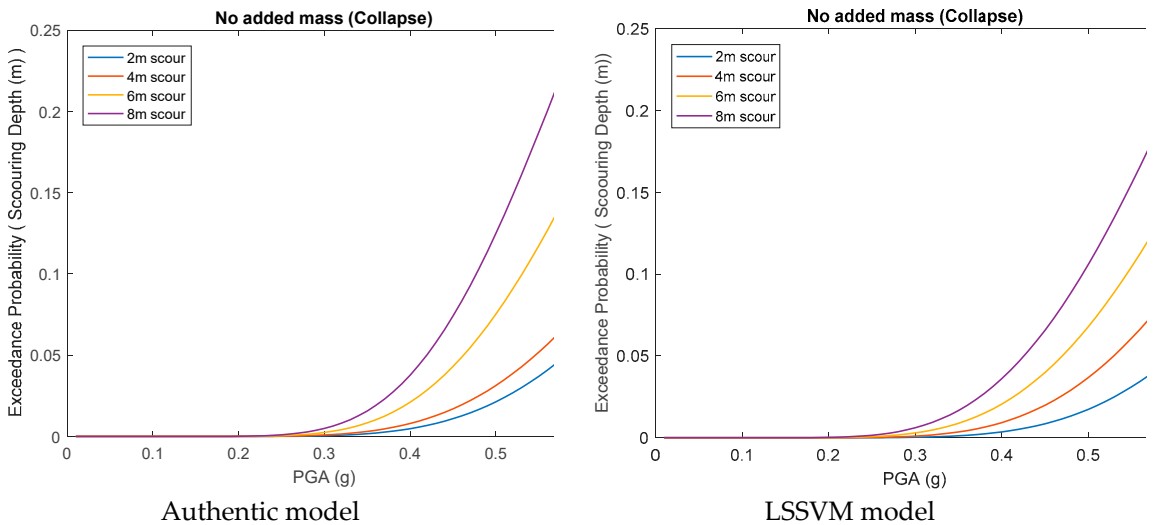

**Figure 20.** Fragility curves using authentic and LSSVM models.

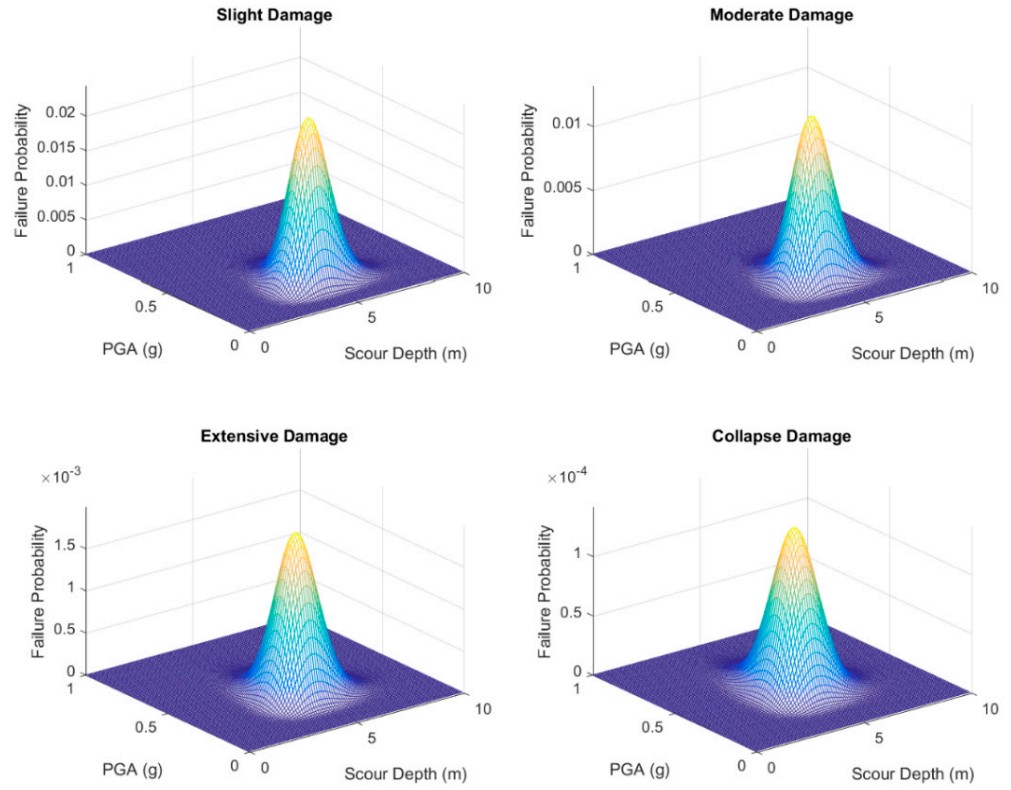

**Figure 21.** Joint failure probability under 4 m immersion depth against floods with a 100 year return period.

## 9. Conclusions

In view that Taiwan is affected by typhoons and earthquakes all year long, this research proposes a process for analyzing immersed river-crossing bridges when subjecting to multiple hazards. In addition to the immersing effect, scouring depth and the seismic hazard are included in failure probability calculations. According to the research results, the following important conclusions are drawn:

I.    When calculating the added mass inferred from Morison equation, the $C_m$ and $C_d$ coefficients are from the semiempirical formula, and both will be affected by the Reynolds number and

II. stream velocity. Because the velocity employed in this research was not so fast, it can be assumed as a constant.

II. Based on the variation of different return years, the scouring depth increases along with the increase of return year.

III. The immersing effect is not so apparent as expected when assessing its impact to the safety of a river-crossing bridge, and if the pier is not long enough, the immersing effect may not be so prominent. However, the influence of immerse effect on the bridge is on the non-conservation side, it is important to know that the immersing effect should not be ignored when the bridge pier is long.

IV. This research uses the displacement ductility (*R*) as the performance measurement, and such displacement ductility is the ratio between ultimate displacement and yielding displacement. It is seen that the yielding displacement plays a significant role in safety measurements. In light of this, this study proposes a way to calculate the yielding displacement.

V. The error indicator of LSSVM is very ideal, and the trend of fragility curves drafted according to such results is very close to that established with the authentic model. If computation time is an issue, LSSVM provides an alternative to shorten the analysis time for fragility analyses without scarifying the accuracy.

**Author Contributions:** Conceptualization, K.W. and F.S.; methodology, K.W. and F.S.; software, F.S. and R.J.; validation, K.W., F.S. and R.J.; formal analysis, F.S. and R.J.; investigation, K.W. and R.S.; resources, K.W.; data curation, F.S.; writing—original draft preparation, K.W. and F.S.; writing—review and editing, K.W.; visualization, R.J.

**Funding:** This study was supported by the Ministry of Science and Technology of Taiwan under grant number 107-2622-E-011-020-CC2.

**Conflicts of Interest:** The authors declare no conflict of interest.

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
