# Peer review of "Safety Evaluation of a Water-Immersed Bridge Against Multiple Hazards via Machine Learning"

_applsci, doi:10.3390/app9153116_

Round 1
Reviewer 1 Report
The authors propose a safety evaluation approach for a water-immersed bridge using fragility curves based on a machine learning technique (LSSVM). This approach can be more efficient than the traditional approach with the authentic model. The topic seems timely and of engineering value.
The process of approaches was logically written, but there are some major and minor issues as following:
Major Issues
1. There is no literature review of existing safety evaluation approaches. Previous studies for safety evaluation must be reviewed and discussed while arguing the originality and applicability of this study.
2. Without the appropriate literature reviews, the paper does not show a sound problem statement. Are the existing methods really problematic? What are the significant limitations of them?
3. Abstract and Introduction
1) [Lines 12-13, 37] Authors said that existing safety evaluation approach using authentic model needs much time and many efforts, thus making inefficiency for the analysis of multi-hazard such as seismic excitation. However, the purpose of the proposed safety evaluation approach is to find out the exceedance and occurrence probability of the PGA according to the multi-hazard and the degrees of scouring of the water-immersed bridge. From the viewpoint of safety evaluation using the existing authentic model, it seems slow and inefficiency, but more accurate safety evaluation could be possible. If the safety evaluation results are not changing every hour or day, isn’t it better to use the correct one even if it is just slightly slow?
2) [Line 38] There are so many machine learning methods for nonlinearity, but what is the reason authors chose the LSSVM?
4. Results and discussions
1) [Lines 325-329] Are there any results of analyzing the effects of water depth using an authentic model for comparing to the results of LSSVM? Are they Figure 16?
The accuracy of the safety evaluation cannot be judged to be accurate only with the similarity of each probability range showing in Figures 16 and 19.
In addition, even if Figure 16 was obtained using authentic model, the range of scouring depths in figure 16, 19 is 4-8, 3.25-7.5, respectively so that the comparison does not seem to be correct. Are there any comparison results between the same scouring depth ranges?
They could give clearer comparison results.
2) [Results and discussions] It was mentioned that ineffectiveness of safety evaluation with the authentic model could be a problem in Introduction and Abstract, how effective is the proposed approach than the authentic model? More details of the effectiveness of machine learning for safety evaluation need to be carried out.
5. Minor issues,
1) In Figure 1, some abbreviations have been used such as PSO, ED, SC, and EQ. It needs to explain these abbreviations briefly to clarify Figure 1.
2) Table 1 needs to be explained in detail.
Author Response
Reviewer #1:
The authors propose a safety evaluation approach for a water-immersed bridge using fragility curves based on a machine learning technique (LSSVM). This approach can be more efficient than the traditional approach with the authentic model. The topic seems timely and of engineering value.
The process of approaches was logically written, but there are some major and minor issues as following:
Response:
Many thanks for reviewer’s suggestions and recommendations, which are carefully examined and the corresponding responses are described below.
Major Issues
1. There is no literature review of existing safety evaluation approaches. Previous studies for safety evaluation must be reviewed and discussed while arguing the originality and applicability of this study.
Response:
Many thanks for the suggestion. Related literature review has been added in the revised manuscript. For details, please refer to the “Introduction” Section.
2. Without the appropriate literature reviews, the paper does not show a sound problem statement. Are the existing methods really problematic? What are the significant limitations of them?
Response:
One of the main goals of this study is to extend the existing method to include the immersed effect in bridge safety evaluation. That is, this study does not intend to emphasize the drawbacks in current approaches with respect to the methodology. As for the efficiency for the existing method, this study proposes to use the LSSVM to reduce the calculation effort. Authors have clarified the above statement in the revised manuscript (the section of “Introduction”).
3. Abstract and Introduction
1) [Lines 12-13, 37] Authors said that existing safety evaluation approach using authentic model needs much time and many efforts, thus making inefficiency for the analysis of multi-hazard such as seismic excitation. However, the purpose of the proposed safety evaluation approach is to find out the exceedance and occurrence probability of the PGA according to the multi-hazard and the degrees of scouring of the water-immersed bridge. From the viewpoint of safety evaluation using the existing authentic model, it seems slow and inefficiency, but more accurate safety evaluation could be possible. If the safety evaluation results are not changing every hour or day, isn’t it better to use the correct one even if it is just slightly slow?
Response:
Many pre-analyses such as material nonlinear properties, added mass, soil spring, plastic hinge length and property, nonlinear pushover analysis and time history analysis are needed to build the fragility curve. Taking nonlinear time history analysis for example, for each seismic intensity (e.g., the 475 return period), 7 analyses are needed. Such analyses often need several days or weeks to complete. Yes, authors agree to the reviewer’s concern, if time is not a barrier for a project, one should use the authentic model for safety evaluation The conclusion of the manuscript is revised accordingly as shown below.
“The error indicator of LSSVM Model is very ideal, and the trend of fragility curve drafted according to such result is very close to that established with the authentic model. If computation time is an issue, LSSVM provides an alternative to shorten the analysis time for the fragility analysis without scarifying the accuracy.”
2) [Line 38] There are so many machine learning methods for nonlinearity, but what is the reason authors chose the LSSVM?
Response:
Many thanks for the suggestion. The manuscript is revised accordingly as shown below.
“Please note that there are many available machine learning algorithms, LSSVM is an open source with possibility of any extension in application. Therefore, LSSVM is adopted here. Investigation or comparison with other algorithms is beyond the scope of the current study.”
4. Results and discussions
1) [Lines 325-329] Are there any results of analyzing the effects of water depth using an authentic model for comparing to the results of LSSVM? Are they Figure 16?
The accuracy of the safety evaluation cannot be judged to be accurate only with the similarity of each probability range showing in Figures 16 and 19.
In addition, even if Figure 16 was obtained using authentic model, the range of scouring depths in figure 16, 19 is 4-8, 3.25-7.5, respectively so that the comparison does not seem to be correct. Are there any comparison results between the same scouring depth ranges?
They could give clearer comparison results.
Response:
Many thanks for the suggestion. As suggested, the direct comparison of using two different approaches are included in the revised paper as shown below. It is seen that the trends of two approaches are similar. The greater the scour depth, the bigger the difference.
Authentic model | LSSVM model |
2) [Results and discussions] It was mentioned that ineffectiveness of safety evaluation with the authentic model could be a problem in Introduction and Abstract, how effective is the proposed approach than the authentic model? More details of the effectiveness of machine learning for safety evaluation need to be carried out.
Response:
The effectiveness of the proposed approach depends on the complexity of a given bridge. If the authentic approach is used, many pre-analyses such as material nonlinear properties, added mass, soil spring, plastic hinge length and property, nonlinear pushover analysis and time history analysis are needed to build the fragility curve. Taking nonlinear time history analysis in this study for example, for each seismic intensity (e.g., the 475 return period), 7 analyses are needed. Such analyses often need several days or weeks to complete. If LSSVM is adopted, one can acquire the displacement within a few minutes.
The above description is included in the revised manuscript.
5. Minor issues,
1) In Figure 1, some abbreviations have been used such as PSO, ED, SC, and EQ. It needs to explain these abbreviations briefly to clarify Figure 1.
Response:
The explanation of each term is provided in the revised paper.
2) Table 1 needs to be explained in detail.
Response:
The following description is added in the revised paper.
“Indicated in Table 1 is the result of errors and relative coefficient obtained during the LSSVM training for the case of scour depth is 0m. It is shown that the prediction performance is not perfect, but it is acceptable. To be more specific, Lewis (1982) suggested that if the MAPE is less than 10%, the prediction performance is considered as highly accurate. If the MAPE is between 11% to 20%, the prediction performance is good.”
Reviewer 2 Report
The article submitted presents an analytical method to analyse the safety of an immersed bridge in Taiwan. The method includes nonlinear FEM analysis complemented with machine learning methodology to obtain the fragility curves of the bridge, allowing to reduce the computational effort. For this, some parameters were varied, namely the water depth, scouring depth and seismic intensity.
The bridge modelling and the used methodology to analyse the structure is explained. The obtained results are discussed. It is shown that the proposed methodology is suitable to assess the safety of the bridge under multi-hazards.
The topic that is developed in the article is very interesting and actual. It shows how recent developments on computing science can be used to complement traditional structural analysis methods. The proposed methodology is suitable to be applied to other cases of practice and can be very useful for structural engineers and political decision-makers.
I made few specific comments (see below) to improve the article. The reviewer encourages the authors to take the few suggestions into account and resubmit a reviewed version of the article.
Comment 1
The entire article must be reviewed to improve the reading. Many sentences must be rewritten. I present only one example, but they exist many along the text: Page 2, line 68: the sentence “The sectional diameter represents and linear interpolation value …” has no sense.
Comment 2
Several typos throughout the article must be corrected. I give some examples but the article must be carefully reviewed.
- In Figure 3, the units for the strain are wrong,
- In page 7, line 154, the reference is wrong, it should be “Melville and Coleman (2000)” Please check! Same in Page 8, line 162);
- Some equations must be rescaled. For instance Eq. (12) is too small (please correct ()(12));
- Page 9, line 194/195, It should be “the ratio between the ultimate displacement and yielding displacement…”
- …
Comment 3
In section 7, I didn´t find how the concept of LSSVM was implemented in computer or which software was used. Please add a short note to explain this.
Author Response
Reviewer #2:
The article submitted presents an analytical method to analyse the safety of an immersed bridge in Taiwan. The method includes nonlinear FEM analysis complemented with machine learning methodology to obtain the fragility curves of the bridge, allowing to reduce the computational effort. For this, some parameters were varied, namely the water depth, scouring depth and seismic intensity.
The bridge modelling and the used methodology to analyse the structure is explained. The obtained results are discussed. It is shown that the proposed methodology is suitable to assess the safety of the bridge under multi-hazards.
The topic that is developed in the article is very interesting and actual. It shows how recent developments on computing science can be used to complement traditional structural analysis methods. The proposed methodology is suitable to be applied to other cases of practice and can be very useful for structural engineers and political decision-makers.
I made few specific comments (see below) to improve the article. The reviewer encourages the authors to take the few suggestions into account and resubmit a reviewed version of the article.
Response:
Many thanks for reviewer’s suggestions and recommendations, which are carefully examined and the corresponding responses are described below.
Comment 1
The entire article must be reviewed to improve the reading. Many sentences must be rewritten. I present only one example, but they exist many along the text: Page 2, line 68: the sentence “The sectional diameter represents and linear interpolation value …” has no sense.
Response:
Many thanks for the recommendations. The sentence has been corrected and the entire manuscript is carefully reviewed before resubmitting.
Comment 2
Several typos throughout the article must be corrected. I give some examples but the article must be carefully reviewed.
- In Figure 3, the units for the strain are wrong,
Response:
Many thanks for the reminder. Figure 3 is corrected accordingly.
- In page 7, line 154, the reference is wrong, it should be “Melville and Coleman (2000)” Please check! Same in Page 8, line 162);
Response:
Many thanks for the reminder. Literature are corrected accordingly.
- Some equations must be rescaled. For instance Eq. (12) is too small (please correct ()(12));
Response:
Many thanks for the reminder. All equations are reviewed and enlarged if it is too small.
- Page 9, line 194/195, It should be “the ratio between the ultimate displacement and yielding displacement…”
Response:
Many thanks for the reminder. The definition of displacement ductility is revised accordingly.
Comment 3
In section 7, I didn´t find how the concept of LSSVM was implemented in computer or which software was used. Please add a short note to explain this.
Response:
The following description is included in the revised paper.
“To implement the LSSVM in computer, this study utilizes the LSSVMlab toolbox provided by Suykens et al. (2002), which contains Matlab/C implementations for a number of LSSVM algorithms.”
Round 2
Reviewer 1 Report
While many issues have been addressed, there are still major and minor concerns to be addressed before the paper can be accepted for publication.
Major Issues
1. As reviewer has mentioned before in the previous version, there are many nonlinear machine learning methods such as lasso regression, Bayesian inference, deep learning (RNN, DNN), SVM and so on. Meanwhile, each method may have a specific domain or specific parts of an analysis.
Please clarify the following questions.
1) In terms of data the authors have, what is the superiority of LSSVM over other machine learning methods?
2) Are there other studies using the LSSVM for safety evaluation?
3) If not, how was LSSVM used in other domains and how accurate the prediction using LSSVM is?
The answers to these questions are critical to justify the author’s opinion for using LSSVM, and can be a great help to other researchers considering the use of LSSVM.
2. There are some issues in Figure 20. First, the legend of Figure 20a does not represent all of the lines appropriately.
It can be also helpful for the comparisons between the authentic model and the LSSVM model to make each y-axis of Figures 20a and 20b having the same level.
If what the reviewer understand rightly, the four lines (green, purple, yellow and orange) in Figure 20a mean the scour depth of 8 (green), 6 (purple), 4 (yellow), 2 (orange).
It shows exceedance probability (EP) of LSSVM model has a lower EP than the authentic model overall.
In other words, the LSSVM model overestimates safety more than it actually is. If it is used in the actual field, this can be a critical problem. Therefore, it seems that this problem should be solved. If the EPs of LSSVM model is higher than Eps of authentic model, it would be rather be acceptable in terms of safety.
3. Minor issue,
1) In Line 53, the expression ‘LSSVM is an open source’ is not appropriate. Isn’t it correct to say that LSSVMlab toolbox used by authors is an open source?
Author Response
Applied Sciences
Safety evaluation of a water-immersed bridge against multi-hazards via machine learning
<Manuscript ID: applsci-539402 >
Dear Editor-in-Chief
We deeply appreciated the editors and reviewers' comments and suggestions for our manuscript. The manuscript is revised accordingly. Detailed modifications or explanations are listed below point to point.
Reviewer #1:
While many issues have been addressed, there are still major and minor concerns to be addressed before the paper can be accepted for publication.
Response:
Many thanks for reviewer’s suggestions and recommendations, which are carefully examined and the corresponding responses are described below.
Major Issues
1. As reviewer has mentioned before in the previous version, there are many nonlinear machine learning methods such as lasso regression, Bayesian inference, deep learning (RNN, DNN), SVM and so on. Meanwhile, each method may have a specific domain or specific parts of an analysis.
Please clarify the following questions.
1) In terms of data the authors have, what is the superiority of LSSVM over other machine learning methods?
Response:
“The formulation of LSSVM is flexible and can be easily incorporated with existing/known information/data due to the parametric model in the primal domain. The aforementioned prior knowledge, such as water level and scouring depth considered here, can be automatically embedded in the dual formulation.”
The above statement is included in the revised manuscript.
2) Are there other studies using the LSSVM for safety evaluation?
Response:
Yes, Ji et al. (2016) utilized LSSVM for slope system reliability analysis and Liao et al. (2017) adopted LSSVM for seawall safety evaluation. The above literatures are listed below and are included in the revised manuscript.
Ji, J., Zhang, C., Gui, Y., Lü, Q., & Kodikara, J. (2016). New observations on the application of LS-SVM in slope system reliability analysis. Journal of Computing in Civil Engineering, 31(2), 06016002.
Liao, K. W., & Hsiung, H. J. (2017). Reliability-Based Safety Analysis of a Seawall with Incomplete Information on Tsunamis. Coastal Engineering Journal, 59(03), 1750014.
3) If not, how was LSSVM used in other domains and how accurate the prediction using LSSVM is?
Response:
LSSVM has been successfully applied to many different fields such as pharmaceutical industry (Parhizkar et al. 2109) and vibration signal process (Li et al. 2019) as shown below. The above literatures are are included in the revised manuscript.
Parhizkar, E., Saeedzadeh, H., Ahmadi, F., Ghazali, M., & Sakhteman, A. (2019). Partial least squares-least squares-support vector machine modeling of ATR-IR as a spectrophotometric method for detection and determination of iron in pharmaceutical formulations. Iranian journal of pharmaceutical research: IJPR, 18(1), 72.
Li, X., Yang, Y., Pan, H., Cheng, J., & Cheng, J. (2019). A novel deep stacking least squares support vector machine for rolling bearing fault diagnosis. Computers in Industry, 110, 36-47.
The answers to these questions are critical to justify the author’s opinion for using LSSVM, and can be a great help to other researchers considering the use of LSSVM.
Response:
Many thanks for reviewer’s suggestions and recommendations, the manuscript is revised accordingly as described above.
2. There are some issues in Figure 20. First, the legend of Figure 20a does not represent all of the lines appropriately.
Response:
Many thanks for reviewer’s reminder. The figure is revised accordingly.
It can be also helpful for the comparisons between the authentic model and the LSSVM model to make each y-axis of Figures 20a and 20b having the same level.
Response:
Many thanks for reviewer’s reminder. The figure is revised accordingly.
If what the reviewer understand rightly, the four lines (green, purple, yellow and orange) in Figure 20a mean the scour depth of 8 (green), 6 (purple), 4 (yellow), 2 (orange).
It shows exceedance probability (EP) of LSSVM model has a lower EP than the authentic model overall.
In other words, the LSSVM model overestimates safety more than it actually is. If it is used in the actual field, this can be a critical problem. Therefore, it seems that this problem should be solved. If the EPs of LSSVM model is higher than Eps of authentic model, it would be rather be acceptable in terms of safety.
Response:
Yes, some of the LSSVM estimations have lower EPs than those of the authentic model. The proposed surrogate model is applied to Equation (23) only when the authentic model is not available. That is, if EP from authentic model is unavailable, interpolated data from LSSVM is adopted to build plots in Figure 21. Authors agree with reviewer’s concern, a notice of using interpolation data is provided in the revised manuscript as shown below.
“Please note that the interpolated data between authentic ones may have a higher/lower EP than that of the authentic one, as shown in Figure 20. The proposed surrogate model is applied to Equation (23) only when the authentic model is not available. That is, if EP from authentic model is unavailable, interpolated data from LSSVM is adopted to build plots in Figure 21. Caution should be taken in using results of surrogate model.”
3. Minor issue,
1) In Line 53, the expression ‘LSSVM is an open source’ is not appropriate. Isn’t it correct to say that LSSVMlab toolbox used by authors is an open source?
Response:
Many thanks for reviewer’s reminder. The manuscript is revised accordingly.

Round 3
Reviewer 1 Report
While many issues raised previously have been reasonably dealt with, some issues have not been yet explained sufficiently.
1. It is necessary to explain in detail the effects on safety evaluation when the EPs of the LSSVM model are lower or higher than those of the authentic model, respectively. This issue is about focusing on false positive or false negative. In aspect of FP and FN, how the EPs of the LSSVM should be interpreted when the EPs are higher or lower than those of the authentic model, respectively?
2. What is a possibility that a LSSVM model provides higher or lower EPs than the authentic model, respectively? If the possibility cannot be predicted and controlled, the reliability of the LSSVM model is highly suspicious. The reliability of the proposed LSSVM model should be explained in detail.
Author Response
Applied Sciences
Safety evaluation of a water-immersed bridge against multi-hazards via machine learning
<Manuscript ID: applsci-539402 >
Dear Editor-in-Chief
We deeply appreciated the editors and reviewers' comments and suggestions for our manuscript. The manuscript is revised accordingly. Detailed modifications or explanations are listed below point to point.
Reviewer #1:
While many issues raised previously have been reasonably dealt with, some issues have not been yet explained sufficiently.
Response:
Many thanks for reviewer’s suggestions and recommendations, which are carefully examined and the corresponding responses are described below.
1. It is necessary to explain in detail the effects on safety evaluation when the EPs of the LSSVM model are lower or higher than those of the authentic model, respectively. This issue is about focusing on false positive or false negative. In aspect of FP and FN, how the EPs of the LSSVM should be interpreted when the EPs are higher or lower than those of the authentic model, respectively?
Response:
Many thanks for reviewer’s recommendations. The main purpose of utilizing LSSVM for fragility analysis is to reduce the computational cost and remain the correct trends for engineering preliminary analysis. A failure decision for a given bridge depends not only on the failure probability but also on other factors such as the safety threshold, design PGA, scour depth and performance level. Taking the bridge in Figure 20 for example, based on the bridge site, the design PGA is 0.24g, assuming that the safety threshold is 0.00135 that is equivalent to a common reliability index (i.e., b = 3) used for infrastructure. Under these conditions, comparison of the authentic and predicted fragility curves are displayed below. It is seen that among four different scour depths with collapse performance limit state considered, the case of 8-meter scour can be considered as a false negative event. Thus, if the predicted fragility is used accompanied with other factors such as reliability target, scour depth, and PGA for bridge safety judgement, caution should be taken.
Authentic | predict |
Figure A. Fragility curve comparison of authentic and predict models
2. What is a possibility that a LSSVM model provides higher or lower EPs than the authentic model, respectively? If the possibility cannot be predicted and controlled, the reliability of the LSSVM model is highly suspicious. The reliability of the proposed LSSVM model should be explained in detail.
Response:
Many thanks for reviewer’s recommendations. As shown in Figures 20 and A, the overall trends in these two Figures are different. In Figure 20, the predicted failure probabilities are often greater than those of the authentic ones. In Figure A, the authentic failure probabilities are often greater than those of the predicted ones. It is found that if the extrapolated is adopted when PGA is at a greater level (Figure 20), the predict values tend to be smaller than those of actual ones. On the other hand, if the interpolated is adopted when PGA is at a smaller level (Figure A), the predict values tend to be greater than those of actual ones. This is due to the authentic analyses performed have covered the range of design PGA (it is very unusual to have 0.6g as the design PGA), the interpolation of LSSVM will be adopted at most cases. That is, using LSSVM will provide a conservative estimation of failure probability if the PGA is within the design range.
